# Perceived transcultural self-efficacy and its associated factors among nurses in Ethiopia: A cross-sectional study

**Robera Demissie Berhanu**[1]*, **Abebe Abera Tesema**[2], **Mesfin Beharu Deme**[2], **Shuma Gosha Kanfe**[3]

**1** Department of Nursing, College of Health Sciences, Mettu University, Mettu, Oromia Region, Ethiopia, **2** School of Nursing and Midwifery, Institute of Health Sciences, Jimma University, Jimma, Oromia Region, Ethiopia, **3** Department of Health Informatics, College of Health Sciences, Mettu University, Mettu, Oromia Region, Ethiopia

* roobeeraadb@gmail.com

## Abstract

### Background

Transcultural self-efficacy is a nurse's perception of his or her own ability to accomplish activities effectively for culturally diverse clients. This self-efficacy may be affected by different factors, either positively or negatively. Quality care can be improved significantly when nurses provide patient-centered care that considers cultural background of the patients. Thus, this study aimed to assess perceived transcultural self-efficacy and its associated factors among nurses working at Jimma Medical Center.

### Methods

Facility-based cross-sectional study with both quantitative and qualitative methods of data collection was conducted among 244 nurses and 10 key informants from 20 May to 20 June 2020. Bivariate and multivariable linear regression analyses were used to identify factors associated with transcultural self-efficacy. Qualitative data were coded and analyzed thematically. Quantitative results were integrated with qualitative results.

### Results

A total of 236 nurses participated in the study making the response rate 96.7%. The mean transcultural self-efficacy score was 2.89 ± 0.59. Sex, work experience, intercultural communication, cultural sensitivity, interpersonal communication, and cultural motivation were significantly associated with transcultural self-efficacy. Ten in-depth interviews were conducted and the findings of qualitative data yielded four major themes.

### Conclusion

The level of perceived transcultural self-efficacy was moderate among nurses. Transcultural self-efficacy of nurses varies with several factors including sex, experience, intercultural

**Data Availability Statement:** All the data underlying this study are provided in the Supporting information file.

**Funding:** The authors received no specific funding for this work.

**Competing interests:** The authors declare that they have no competing interests.

**Abbreviations:** CI, Confidence interval; CMS, Cultural motivation scale; CSES, Cultural self-efficacy scale; CSS, Cultural sensitivity scale; ETB, Ethiopian birr; ICS, Intercultural communication scale; IDI, In-depth interview; IPCS, Interpersonal communication scale; JMC, Jimma medical center; M, Mean; SD, Standard deviation; SE, Standard error; TNT, Transcultural nursing theory; TSE, Transcultural self-efficacy; VIF, Variance inflation factor.

communication, cultural sensitivity, interpersonal communication, and cultural motivation. This calls for the need to offer transcultural nursing training for nurses.

## Introduction

Self-efficacy is one's own belief in one's ability to succeed in specific situations or accomplish a task as defined by psychologist Albert Bandura. One's self-efficacy perceptions can have an important role in how one deals with challenging situations [1]. Transcultural self-efficacy (TSE) is defined as the perception or expectation of a nurse regarding his or her own ability to accomplish activities effectively in the conditions characterized by cultural diversity [2]. It is a nurse's confidence in delivering care to patients coming from different cultural backgrounds or it indicates nurses' degree of confidence to provide both culture-general and culture-specific care [3,4].

The history of culture is as old as the history of human beings but it was during the time of Madeline Leininger that culture became the central concept in the transcultural nursing theory (TNT) and evidence-based area of study. The goal of the theory is to develop a body of knowledge to establish nursing care that is sometimes culture-specific and sometimes culture-universal. This helps researchers and clinicians discover and explain the interdependence of care and culture while understanding cultural similarities and differences [5,6]. In our today's world, globalization has resulted in a new way to approach clients considering their cultural background. The ability to recognize and embrace cultural diversity is of utmost importance to all health care providers, particularly to nurses since they come into very intimate contact with health care seekers of different cultures. Nurses are required to understand the cultural care beliefs, values, and lifeways of patients and then they need to immerse themselves in the culture of all patients to deliver care that is parallel to what the patient believes suitable according to his or her cultural expectations [7–9]. Therefore, nurses must be prepared to recognize clients' desires that are derived from their culture and to develop skills that will help facilitate their achievement [10].

Literature reveals that culture is central to the human condition, and nurses who are transculturally confident in understandings of health and disease held by patients are needed for today's multicultural population. Quality care can be improved significantly when nurses provide patient-centered care that considers the cultural background of the patients. Accordingly, nurses should possess the cultural knowledge required to deliver effective and satisfactory care that is congruent with the culture of the patient [11–14].

Nursing is a profession that relies on a holistic approach to delivering health care. This includes taking into account patients' cultural needs [11]. Globally, health care is becoming more culturally and ethnically diverse over the last 20 years [15]. Diversity has increased in many countries due to wars, discrimination, political strife, and worldwide socioeconomic conditions. There are also national concerns about health disparities since the cultural diversity of the population is increasing. Numerous studies indicate that health disparities continue to demonstrate evidence for health care providers to pay attention to cultural diversity [16–18]. Since nurses lie at the front line of health care providers, they have the opportunity to stand out as the ones who can understand and are sensitive to the cultures of patients by helping patients with their particular lifeways and in their environmental context [9,19].

However, researchers identified that working with culturally diverse patients can be challenging due to differences in ideas, beliefs, thoughts, norms, customs, and traditions. If poorly

managed, this can result in miscommunication, maladaptive behaviors, and interpersonal conflicts [15,20–22]. Even if nurses are educated in Ethiopia with the curriculum adapted from the international curriculum, they provide care for patients unique in culture from those of patients where transcultural self-efficacy is well studied. Therefore, the finding from this study is useful to provide evidence-based nursing care to culturally and ethnically diverse clients. Thus, this study aims to assess perceived transcultural self-efficacy and its associated factors among nurses working at Jimma Medical Center.

## Materials and methods

### Study design

This study employed a cross-sectional study design with a mixed method of data collection.

### Study site, study population, and recruitment criteria

This study was conducted at Jimma Medical Center (JMC). Geographically, it is located in Jimma town which is found at 352km from Addis Ababa, the Capital city of Ethiopia. It has outpatient and inpatient services, maternal and child health services, referral and follow-up services, physiotherapy and rehabilitative services, intensive care, and recovery services. Approximately, it provides service for 15,000 inpatients, 160,000 outpatient attendants, 11,000 emergency cases, and 4500 deliveries in a year coming to the hospital from the catchment population of about 15 to 20 million people. The hospital had 628 nurses and midwives currently. Among these, 553 were diploma nurses, B.Sc. nurses, and M.Sc. nurses. Out of these, 518 nurses were currently working at JMC while 35 attending education. The study was conducted from 20 May to 20 June 2020.

The study population for this study was all nurses working at JMC who fulfilled eligibility criteria. Nurses who had at least 6-month work experience at JMC were included in the quantitative strand of the study. Nurses who had greater than 2-years of work experience were included purposely in the qualitative strand of the study for they could provide detailed information. Nurses who were not found during the data collection period due to annual leave, maternal leave, or some other social problems were excluded.

### Sample size determination

Quantitative sample size was calculated using a formula for a single population proportion considering confidence interval of 95% (Z = 1.96), margin of error 5% (d = 0.05), and population proportion of 50% (p = 0.5) since there is no study on transcultural self-efficacy of nurses or predictor of transcultural self-efficacy in Ethiopia. The initial calculated sample size was 384. Since the total number of nurses in JMC is 518, which is less than 10,000; and also since the ratio of initial sample size calculated to the total number of nurses in JMC (N) is greater than 0.05, correction formula was used and the minimum sample size required became 221. The final sample size was 244 after 10% non-response was added.

The total number of in-depth interviews was determined by the level of data saturation and a minimum of eight in-depth interviews were conducted.

### Sampling and data collection

Simple random sampling technique was used to recruit participants in the quantitative part of the study. First, the list of all nurses was taken from the matron office of JMC with their corresponding wards or departments. This list was used as a sampling frame. Then, simple random sampling using lottery methods was employed to select the nurses from the hospital. Purposive

sampling technique was used to recruit participants for in-depth interviews. Qualitative data were collected based on specified criteria throughout the interviews. Participants were willing and who had greater than 2 years of experience were selected for in-depth interviews. Four B. Sc. nurses were recruited from Agaro General Hospital for facilitating quantitative data collection. Continuous monitoring was made by the principal investigators during data collection. Before actual data collection, the data collection tools were pretested on 25 nurses working at Agaro General Hospital to measure the internal consistency of the tools, to estimate data collection time, and to make modifications as needed. The internal consistency was tested using a reliability scale. Qualitative data were collected by the principal investigators.

## Operational definitions

**Transcultural self-efficacy.**   Nurse's confidence when providing care for patients from different cultural backgrounds measured using 25 items on a 5-point Likert scale with the value ranging from 25–125. The higher the value, the higher the confidence of nurses in providing care to patients of different cultural backgrounds. The overall mean score of TSE was divided by 25 (the total number of items) giving the value ranging from 1 to 5. Then TSE level was divided into low, moderate, and high based on this score. A low level is a mean score between 1 and 2.33, a moderate level is a mean score between 2.34 and 3.67, and a high level is a mean score between 3.68 and 5 [23].

**Intercultural communication.**   Nurses' communication effectiveness when caring for patients from different cultural backgrounds measured using 5 items on a 5-point Likert scale with the value ranging from 5–25.

**Cultural sensitivity.**   Knowledge, awareness, and acceptance of other cultures measured using 3 items on a Likert scale with the value ranging from 3–15.

**Interpersonal communication.**   Nurse's communication effectiveness between people measured using 7 items on a 5-point Likert scale with the value ranging from 7–35.

**Cultural motivation.**   Nurse's actions, desires, and needs to learn about and engage with culture-based care measured using 5 items on a Likert scale with the value ranging from 5–25.

## Study instruments

A structured self-administered questionnaire was used to collect data for this study. The questionnaire had six parts: (i) socio-demographic questionnaire, (ii) cultural self-efficacy scale (CSES), (iii) intercultural communication scale (ICS), (iv) cultural sensitivity scale (CSS), (v) interpersonal communication scale (IPCS) and (vi) cultural motivation scale (CMS). The questionnaire was adapted from related studies and was used for data collection [24–27]. It was administered in English.

The CSES consists of three subscales. The CSES-Knowledge scale consists of 16 items that ask about knowledge of many health-related domains (e.g., health practices, nutritional patterns, employment patterns). The CSES-Skills scale consists of six items and asks about self-efficacy in performing various skills, such as advocacy and entering an ethnically distinct community. The CSES-Cultural Concepts scale consists of three items and assesses knowledge of cultural concepts (i.e., ability to distinguish inter- from intracultural diversity, ability to distinguish ethnocentrism from discrimination, ability to distinguish between ethnicity and culture). Ratings on all items are made on a 1 to 5 scale standing for the words "very little confidence" for a score of 1, "little confidence" for a score of 2, neutral or noncommittal confidence for a score of 3, "moderate confidence" for a score of 4 and "high confidence" for a score of 5. The total scale score is between 25 and 125 in all subscales. The Cronbach's alpha was 0.97 when it was developed [28] and Herrero Hahn et al. (2017) reported Cronbach's alpha of 0.978 [3].

The ICS consists of five questions that ask nurses to think about how they feel when communicating with patients from different cultural backgrounds from their own and rate their ability to understand feelings, communicate, resolve misunderstandings, comprehend points of view, and empathize with patients. Items were rated using a 5-point Likert-type scale with responses of strongly disagree, disagree, neutral, agree, and strongly agree. The total scale score is between 5 and 25.

CSS consists of 3 items that ask to know about patients' cultures, adapting treatments for patients, and considering culture when making recommendations. The scale is measured on a 5-point Likert scale with responses ranging from strongly disagree to strongly agree. Ulrey and Amason (2001) reported Cronbach's alpha of 0.83 for ICS [27]. The IPCS was used to measure interpersonal communication. The adapted scale consists of 7 items measured with responses ranging from strongly disagree to strongly disagree. Asurakkody (2019) reported Cronbach's alpha of 0.75 for IPCS [29]. Cultural motivation was measured by the cultural motivation scale (CMS) adapted from 5 items subscale of the cultural intelligence scale. The scale consists of Likert-type questions with responses ranging from strongly disagree to strongly agree. The total score of CMS is between 5 and 25. Its' Cronbach's alpha was 0.75 [29].

Semi-structured in-depth interview (IDI) guiding questions were used. The questions had two parts: socio-demographic characteristics related questions and questions regarding transcultural self-efficacy. The IDI guiding questions were checked by experts. The IDIs were conducted in the silent and comfortable room (home, workplace) after time and place were decided by the interviewees. The interviews were conducted face to face in the well-ventilated rooms after both interviewers and interviewees sat at 2 meters distance wearing face mask. The principal investigators (RDB and MBD) conducted the interviews in Afaan Oromo and Amharic. Field notes were taken and a tape recorder was used to record the interviews lasting 20–25 minutes.

## Data quality management

**Quantitative part.**   Pretest was conducted on 25 (10%) nurses at Agaro General Hospital. The reliability of the data collection tool was measured, data collection time was estimated and some modifications such as logical order and rewriting items difficult to understand were made as well. One-day training was given for data collection facilitators regarding the objective of the study, data collection tools and procedures, how to approach respondents, and how to keep confidentiality. Finally, the collected data were checked for completeness.

**Qualitative part.**   Credibility was assured through long-lasting engagement with IDI participants and by investing sufficient time to become familiar with the setting and context, to test for misinformation, to build trust, and to get to know the data to get rich. Participants who were genuinely willing to take part and prepared to offer data freely were involved in the study. Transferability was assured by describing the behavior and experiences of the participants with their context so that the behavior and experiences become meaningful to an outsider. Dependability and confirmability were assured through transparently describing the research steps taken from the start of a research proposal to the development and reporting of the findings. The records of the research path were kept throughout the study.

## Data analysis

**Quantitative part.**   Data were analyzed using IBM SPSS Statistics Version 26. Then data were analyzed using descriptive statistics such as mean scores and standard deviations followed by inferential statistics. Mean scores and standard deviations were computed to determine the level of transcultural self-efficacy of nurses. All variables were entered into bivariate

linear regression analysis and independent variables with $p < .25$ were considered candidates for multivariable linear regression analysis. In the multivariable linear regression model, statistical significance was declared at $p < .05$. Linear regression assumptions were checked before data analysis. Normality assumption was checked by Kolmogorov-Smirnov test, which was insignificant. The result showed the data were approximately normally distributed. Collinearity assumption was checked to see the correlation between independent variables by using the variance inflation factor (VIF). The results showed that all the variables had VIF less than 5 (Minimum VIF = 1.238 and Maximum VIF = 3.580).

**Qualitative part.** Interviews were transcribed verbatim into plain text, translated into English, and then imported directly into ATLAS.ti. 7.5.16 for coding and analyzed thematically. Accordingly, each text was read again and again carefully and then codes were generated by two data coders. Data relevant to each code were collected together. Then codes were grouped through constant comparison of concepts, like with like, into four major themes. It was checked whether the themes work with the coded extracts.

## Ethics approval and consent to participate

The study was reviewed and approved by Jimma University's Ethical Institutional Review Board.

Permission was obtained from JMC where the study was carried out. All participants were provided written informed consent to participate in the study.

## Results

Two hundred thirty-six (236) of the 244 invited participants completed the questionnaires making the response rate 96.7%. Additionally, in-depth interview was conducted among 10 nurses. The interview findings were finalized into four major themes: transcultural self-efficacy perception, sociodemographic factors that influence transcultural self-efficacy, cultural factors influencing transcultural self-efficacy, and person-to-person communication skills.

## Socio-demographic characteristics

The mean age of the respondents was 27.53 years (SD = ±3.13 years), majority 177(75.0%) of them were in the age group of 25–29 years. More than half 132 (55.9%) of them were female and 138(58.5%) of them were Oromo. One hundred one (42.8%) of the respondents were orthodox in religion. More than three fourth 190(80.5%) of them had B.Sc. degree in nursing. Nearly half 115(48.7%) of them had monthly income which falls in the range of 4,001.00–6,000.00 ETB. Regarding the length of experience, more than two-third 162(68.6%) of them had 2–4 years of experience [Table 1].

## Transcultural self-efficacy level

Descriptive statistics were performed to determine the mean and standard deviation of transcultural self-efficacy. The mean TSE score was 2.89 (SD = ± 0.59) out of the total possible scores ranging from 1 to 5 which indicates a moderate level of transcultural self-efficacy. The mean TSE scores for the knowledge of cultural concepts and specific nursing skills were 3.35 (SD = ± 0.78) and 3.14 (SD = ± 0.88) respectively. The lowest mean score was observed in the knowledge of cultural patterns, which was 2.71 (SD = ± 0.61). Therefore, this study result shows that respondents perceived themselves as having moderate transcultural self-efficacy in working with patients from different cultural backgrounds [Table 2]. This is corroborated by the qualitative finding as one 28-year-old male participant said, "*Patients come from different*

**Table 1. Socio-demographic characteristics of nurses working at JMC, Oromia Region, Southwest Ethiopia, May to June 2020 (n = 236).**

| Variables | Categories | Frequency | Percentage (%) |
|---|---|---|---|
| **Age (years)** | <25 | 17 | 7.2 |
| M = 27.53 years | 25–29 | 177 | 75.0 |
| SD = ±3.13 years | 30–34 | 36 | 15.3 |
| | ≥35 | 6 | 2.5 |
| **Sex** | Male | 104 | 44.1 |
| | Female | 132 | 55.9 |
| **Ethnicity** | Oromo | 138 | 58.5 |
| | Amhara | 77 | 32.6 |
| | Dawro | 12 | 5.1 |
| | Other* | 9 | 3.8 |
| **Religion** | Orthodox | 101 | 42.8 |
| | Muslim | 66 | 28.0 |
| | Protestant | 68 | 28.8 |
| | Other$ | 1 | .4 |
| **Level of education** | Diploma | 40 | 16.9 |
| | B.Sc. and above | 196 | 83.1 |
| **Monthly income** | 2,000–4,000 ETB | 46 | 19.5 |
| | 4,001–6,000 ETB | 115 | 48.7 |
| | 6,001–8,000 ETB | 63 | 26.7 |
| | ≥ 8,001 ETB | 12 | 5.1 |
| **Years of nursing experience** | <2 years | 29 | 12.3 |
| | 2–4 years | 162 | 68.6 |
| | 5–9 years | 30 | 12.7 |
| | 10–14 years | 11 | 4.7 |
| | ≥ 15 years | 4 | 1.7 |
| **Current role** | Supervisor nurse | 5 | 2.1 |
| | Head nurse | 8 | 3.4 |
| | Staff nurse | 223 | 94.5 |

*Gurage, Kafa, Sidama, Wolayta, Tigrie.

$Wakeffata.

places, like southern region, Gambela, and Sudan. However, I individually have the confidence of medium type though there are some disagreements among some few things" (**Participant 1**). A 26-year-old female participant also said, "*I imagine it to be medium because there are different cultures and different ways of thinking. Thus if you do not accept this, later it would influence your confidence. . .*"(**Participant 5**).

**Table 2. Descriptive statistics for each subscale of transcultural self-efficacy and the total TSE scale score of nurses working at JMC, Oromia Region, Ethiopia, May to June 2020 (n = 236).**

| Variables | Possible Scores | Mean (Standard deviation) | Minimum | Maximum |
|---|---|---|---|---|
| Cultural concepts scale | 1–5 | 3.35 (SD = ± 0.78) | 2.00 | 5.00 |
| Cultural patterns scale | 1–5 | 2.71 (SD = ± 0.61) | 1.45 | 4.42 |
| Cultural skills scale | 1–5 | 3.14 (SD = ± 0.88) | 1.17 | 5.00 |
| Total TSE scale score | 1–5 | 2.89 (SD = ± 0.59 | 1.65 | 4.67 |

## Factors associated with transcultural self-efficacy

Both bivariate and multivariable linear regression analyses were carried out to identify factors associated with transcultural self-efficacy. Bivariate analysis revealed that age, sex, level of education, monthly income, years of nursing experience, intercultural communication, cultural sensitivity, interpersonal communication, and cultural motivation were significantly associated with transcultural self-efficacy at $p < .25$.

All the independent variables with $p < .25$ in the bivariate linear regression analysis were entered into the multivariable linear regression analysis to identify factors associated with transcultural self-efficacy. Backward elimination was used for selecting variables in the final model. The study results showed that sex, years of nursing experience, intercultural communication, cultural sensitivity, interpersonal communication, and cultural motivation were significantly associated with transcultural self-efficacy. Accordingly, being male increases transcultural self-efficacy by 2.034 times than female ($\beta = 2.034$, $p = .045$). One year increase in years of nursing experience results in 2.096 units increases in transcultural self-efficacy ($\beta = 2.096$, $p = .001$). This finding was corroborated by the qualitative finding as a 25-year-old female participant said, "*...my confidence in delivering care to patients of different cultures is increased as I stay longer in caring practice*" (**Participant 4**).

One unit increase in intercultural communication results in 1.163 units increases in transcultural self-efficacy ($\beta = 1.163$, $p < .001$). The qualitative finding from a 30-year-old female participant supported this, "*...ability to communicate among patients having different cultures influences confidence for caring*" (**Participant 7**). One unit increase in cultural sensitivity leads to 2.075 units increase in transcultural self-efficacy ($\beta = 2.075$, $p < .001$). The qualitative finding supported this in that a 32-year-old male participant said; ...*as nurses are more and more culturally sensitive, they tend to have higher confidence in giving care to patients coming from different cultures*" (**Participant 6**).

One unit increase in interpersonal communication results in 2.061 units increases in transcultural self-efficacy ($\beta = 2.061$, $p < .001$). The qualitative finding supported this in that one female participant said, "*...when I communicate very well, I can solve the problems quickly. Therefore, having good communication ability is crucial to give care with confidence*" (**Participant 5**). One unit increase in cultural motivation results in 0.635 unit increase in transcultural self-efficacy ($\beta = .635$, $p < .001$). This finding was corroborated by the finding from the qualitative study as one 28-year-old male participant said, "*I am willing and motivated to serve these people. Millions of people come to this hospital. But it is not by separating them by race, ethnicity, tradition, value, and belief that you care... therefore, since I am motivated to deliver care in diversity, my perception of confidence in such condition is very good*" (**Participant 1**). Another 27-year-old male participant also said, "*...for example patients come from different regions speaking different languages and having different cultures. This expands your motivation and lets you to the way to handle such diversity. When you find the way, your confidence is increased*" (**Participant 3**) [Table 3].

## Discussion

This study was carried out to assess perceived transcultural self-efficacy and its associated factors among nurses. The study evaluated how nurses perceive their self-efficacy toward patients from different cultures. The finding indicates the mean score of transcultural self-efficacy perceptions to be 2.89. This is a moderate level of transcultural self-efficacy perception. This finding is higher when compared with the finding from the study conducted by Herrero-Hahn et al. (2019) in Columbia in which the mean score was 2.58 [25]. This difference might be due to differences in socio-cultural characteristics and study settings. However, it is lower when

**Table 3. Multivariable linear regression results showing factors associated with transcultural self-efficacy among nurses working at JMC, Oromia Region, Ethiopia, May to June 2020 (n = 236).**

| Predictor variables | Unstandardized coefficients | | p-value | 95% C.I. | |
|---|---|---|---|---|---|
| | β | SE | | Upper | Lower |
| **Age (in years)** | .098 | .238 | .679 | -.370 | .567 |
| **Sex** | | | | | |
| Female (*Reference*) | | | | | |
| Male | 2.034 | 1.008 | **.045** | .048 | 4.020 |
| **Highest level of education** | | | | | |
| Diploma (*Reference*) | | | | | |
| B.Sc. and above | 1.170 | 1.244 | .348 | -1.282 | 3.621 |
| **Monthly income** | .000 | .000 | .567 | -.001 | .000 |
| **Years of nursing experience** | 2.096 | .283 | **< .001** | 1.539 | 2.653 |
| **Intercultural communication** | 1.163 | .244 | **< .001** | .682 | 1.644 |
| **Cultural sensitivity** | 2.075 | .288 | **< .001** | 1.506 | 2.643 |
| **Interpersonal communication** | 2.061 | .217 | **< .001** | 1.633 | 2.490 |
| **Cultural motivation** | .635 | .174 | **< .001** | .293 | .978 |

For final regression: Adjusted $R^2$ = .917, $p <$ .001.

Dependent variable: Transcultural self-efficacy.

compared to the study conducted by Birnbaum (2012) in which the mean score was 3.41 [30]. The discrepancy may be due to the difference in the socio-cultural characteristics of the participants.

The other finding of this study was the identification of factors that are significantly associated with the transcultural self-efficacy perceptions of nurses. Accordingly, sex, years of nursing experience, intercultural communication, cultural sensitivity, interpersonal communication, and cultural motivation were significantly associated with the transcultural self-efficacy perception.

The finding of the present study indicates that being male increases transcultural self-efficacy. This finding is supported by the study of Herrero-Hahn et al. (2019) in which male nurses demonstrated a higher transcultural self-efficacy score than females [25]. However, it is opposite to the study of Asurakkody (2019), in which female nurses demonstrated a higher transcultural self-efficacy score than males [29]. The discrepancy may be because the vast majority (93.1%) of the participants in their study were females. It is also inconsistent with the study of Li et al. (2016), in which there was no significant gender difference in transcultural self-efficacy perception of nurses [31]. The discrepancy might be because they selected study participants using convenience sampling and the vast majority of their sample were females.

The results of the present study also indicate that longer experience predicted higher transcultural self-efficacy. This is consistent with the social cognitive theory, which proposes successful mastery experiences and vicarious experience increase self-efficacy [32]. This finding is also consistent with the study of Herrero-Hahn et al. (2019), in which longer experience positively predicted transcultural self-efficacy, and with the study of Quine et al. (2012), in which experience predicted higher transcultural self-efficacy [25,33]. It is again consistent with the finding from the study of Li et al. (2016), in which nurses who have worked for 15 years or longer had significantly greater TSE scores when compared with other nurses [31].

The finding of the study indicates that intercultural communication is significantly associated with the transcultural self-efficacy of nurses, i.e. as the communication ability of nurses

increases amidst different cultures, their transcultural self-efficacy perception increase. This is supported by the findings from studies of Chan and Sy (2016) and Quine et al. (2012), in which intercultural communication was positively associated with cultural self-efficacy [33,34]. This finding is also consistent with the social cognitive theory which suggests individuals with greater mastery tend to have higher self-efficacy [35].

The findings of the present study also indicate that cultural sensitivity, interpersonal communication, and cultural motivation were significantly associated with transcultural self-efficacy. Nurses who were more culturally sensitive had higher confidence among clients of different cultural backgrounds. Nurses who had higher communication ability with people had greater transcultural self-efficacy. Nurses who had higher motivation to work amidst cultural diversity demonstrated higher perceptions of transcultural self-efficacy. These findings are supported by the study of Asurakkody (2019), in which cultural sensitivity, interpersonal communication, and cultural motivation positively and significantly predicted TSE [29].

The study used both quantitative and qualitative methods of data collection and this maximizes the reliability of the data. However, the study has several limitations. *First*, the study relied on the self-reported data of nurses that may have introduced a social desirability bias. *Second*, the study is limited to JMC; therefore, the generalization of its findings may not be for other hospitals and health centers. *Finally*, causality could not be established because the study was cross-sectional.

## Conclusion

The level of perceived transcultural self-efficacy was moderate among nurses. The study showed that transcultural self-efficacy varies with sex, experience, intercultural communication, cultural sensitivity, interpersonal communication, and cultural motivation. The in-depth interview results also supported many of these findings. The finding of this study encourages nursing curriculum makers in Ethiopia to incorporate transcultural nursing education into the present curriculum and it calls for the need to offer transcultural nursing training for nurses. The finding of this study highlights that it is advantageous if nurses struggle to improve their cultural understanding and knowledge of transcultural nursing theories. Nurse researchers are encouraged to conduct further study including nurses working at public health centers. Longitudinal study is encouraged to determine cause and effect relationships.

## Supporting information

**S1 File. Data.**
(SAV)

**S2 File. In-depth interview translated.**
(DOCX)

**S3 File. Data collection tools.**
(DOCX)

**S4 File. ISSM COREQ checklist.**
(DOCX)

## Acknowledgments

The authors would like to thank all the nurses who were volunteered to participate in this study. The authors also would like to thank Jimma University health science institute and Jimma medical center for all their help.

## Author Contributions

**Conceptualization:** Robera Demissie Berhanu, Abebe Abera Tesema, Mesfin Beharu Deme, Shuma Gosha Kanfe.

**Data curation:** Robera Demissie Berhanu, Abebe Abera Tesema, Mesfin Beharu Deme, Shuma Gosha Kanfe.

**Formal analysis:** Robera Demissie Berhanu, Abebe Abera Tesema, Mesfin Beharu Deme, Shuma Gosha Kanfe.

**Funding acquisition:** Robera Demissie Berhanu, Abebe Abera Tesema, Mesfin Beharu Deme, Shuma Gosha Kanfe.

**Investigation:** Robera Demissie Berhanu, Abebe Abera Tesema, Mesfin Beharu Deme, Shuma Gosha Kanfe.

**Methodology:** Robera Demissie Berhanu, Abebe Abera Tesema, Mesfin Beharu Deme, Shuma Gosha Kanfe.

**Project administration:** Robera Demissie Berhanu, Abebe Abera Tesema, Mesfin Beharu Deme, Shuma Gosha Kanfe.

**Resources:** Robera Demissie Berhanu, Abebe Abera Tesema, Mesfin Beharu Deme, Shuma Gosha Kanfe.

**Software:** Robera Demissie Berhanu, Abebe Abera Tesema, Mesfin Beharu Deme, Shuma Gosha Kanfe.

**Supervision:** Robera Demissie Berhanu, Abebe Abera Tesema, Mesfin Beharu Deme, Shuma Gosha Kanfe.

**Validation:** Robera Demissie Berhanu, Abebe Abera Tesema, Mesfin Beharu Deme, Shuma Gosha Kanfe.

**Visualization:** Robera Demissie Berhanu, Abebe Abera Tesema, Mesfin Beharu Deme, Shuma Gosha Kanfe.

**Writing – original draft:** Robera Demissie Berhanu, Abebe Abera Tesema, Mesfin Beharu Deme, Shuma Gosha Kanfe.

**Writing – review & editing:** Robera Demissie Berhanu, Abebe Abera Tesema, Mesfin Beharu Deme, Shuma Gosha Kanfe.

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
