## [Decision Letter · Decision Letter 0]

9 Feb 2021

PONE-D-20-35734

Perceived Transcultural Self-Efficacy and Its Predictors among Nurses Working at Jimma Medical Center, Ethiopia: A Cross-sectional Study

PLOS ONE

Dear Dr. Berhanu,

Thank you for submitting your manuscript to PLOS ONE. After careful consideration, we feel that it has merit but does not fully meet PLOS ONE’s publication criteria as it currently stands. Therefore, we invite you to submit a revised version of the manuscript that addresses the points raised during the review process.

We look forward to receiving your revised manuscript.

Kind regards,

Sabeena Jalal, MBBS, MSc, MSc, SM

Academic Editor

PLOS ONE

Journal Requirements:

Furthermore, when reporting the results of qualitative research, we suggest consulting the COREQ guidelines: http://intqhc.oxfordjournals.org/content/19/6/349. In this case, please consider including more information on the number of interviewers, their training and characteristics; and please provide the interview guide used.

3.We note that you have indicated that data from this study are available upon request. PLOS only allows data to be available upon request if there are legal or ethical restrictions on sharing data publicly. For information on unacceptable data access restrictions, please see http://journals.plos.org/plosone/s/data-availability#loc-unacceptable-data-access-restrictions.

4.Thank you for stating the following in the Funding Section of your manuscript:

"This research was funded by Jimma University. All financial supports for data collection,

supervision of the study analysis and interpretation of the data were covered by Jimma University."

 "The funder had no role in study design, data collection and analysis, decision to publish, or preparation of the manuscript."

Additional Editor Comments:

Dear Authors,

The topic of your study is important. It is relevant and timely. However, there are some major revisions that are required.

1. Which checklist did you use for your study? We recommend that authors use the COREQ checklist, or other relevant checklists listed by the Equator Network, such as the SRQR, to ensure complete reporting (http://journals.plos.org/plosone/s/submission-guidelines#loc-qualitative-research).

2. Please also mention in the rationale what or how the findings of the study will be useful.

3. The title is too long. A suggestion: Perceived transcultural self-efficacy and its predictors among nurses in Ethiopia.

4. Please provide detailed description of the sampling strategy, why you choose the method that you have described and why not any other method of sampling, including rationale for the recruitment method.

5. Please provide a discussion of potential sources of bias; and

6. Please provide a discussion of limitations.

Specifics as per Reviewer 1:

Abstract

• In order for the reader to understand “transcultural self-efficacy” please revise the first sentence of the background and describe what “transcultural self-efficacy” is.

• Please clarify in the abstract how efficacy and predictors are linked to the working performance or efficiency of nurses’ duties??

• The abstract has to give following key messages:

o What “transcultural self-efficacy is”

o What is its role in dealing with patients effectively

o What does mean score 2.89 mean?

o The results section has to clearly describe perceived self efficacy and its predictors among nurses

o A holistic but succinct conclusion section mentioning the key finding and a key recommendation

Introduction

• Final para: please also mention in the rationale what or how the findings of the study will be useful.

Discussion

• Please add a para on limitation.

Conclusion

• Please limit the conclusion section to one para and make it succinct

Reviewers' comments:

Reviewer's Responses to Questions

**Comments to the Author**

1. Is the manuscript technically sound, and do the data support the conclusions?

Reviewer #1: Partly

Reviewer #2: No

Reviewer #3: Yes

2. Has the statistical analysis been performed appropriately and rigorously? 

Reviewer #1: Yes

Reviewer #2: No

Reviewer #3: Yes

3. Have the authors made all data underlying the findings in their manuscript fully available?

Reviewer #1: Yes

Reviewer #2: No

Reviewer #3: Yes

4. Is the manuscript presented in an intelligible fashion and written in standard English?

Reviewer #1: Yes

Reviewer #2: Yes

Reviewer #3: Yes

5. Review Comments to the Author

Reviewer #1: The title is too long. A suggestion: Perceived transcultural self-efficacy and its predictors among nurses in Ethiopia

Abstract

• In order for the reader to understand “transcultural self-efficacy” please revise the first sentence of the background and describe what “transcultural self-efficacy” is.

• Please clarify in the abstract how efficacy and predictors are linked to the working performance or efficiency of nurses’ duties??

• The abstract has to give following key messages:

o What “transcultural self-efficacy is”

o What is its role in dealing with patients effectively

o What does mean score 2.89 mean?

o The results section has to clearly describe perceived self efficacy and its predictors among nurses

o A holistic but succinct conclusion section mentioning the key finding and a key recommendation

Introduction

• Final para: please also mention in the rationale what or how the findings of the study will be useful.

Discussion

• Please add a para on limitation.

Conclusion

• Please limit the conclusion section to one para and make it succinct.

Reviewer #2: Concept is a good however overall execution results in significant secretion bias. This produces a aberrant results. Definition of independent and dependent variables are not clear. Furthermore does the sample size represent the population of Ethiopia proportionality. Sample size of patient interviewed was n=10 which is too small to draw any statistical significance. Why has p<0.25 used in the bivariate analysis

Reviewer #3: Dear Authors,

This was a well written paper. The topic is relevant especially due to current COVID situation. There is a need for better understanding of patient concerns. The paper highlight the perspective from the developing country. Data from this study would help in improving and developing better patient care. Subject is explained clearly in the introduction however rationale could be improved. Methodology is adequate with clear explanation of questionnaires used in the study. Results are clearly explained and presented in the tables. Discussion is well written and well supported by similar studies. Overall I would support the paper to be accepted and published.

6. PLOS authors have the option to publish the peer review history of their article (what does this mean?). If published, this will include your full peer review and any attached files.

Reviewer #1: **Yes: **Arshad Altaf

Reviewer #2: No

Reviewer #3: No

---

## [Author Response · Author response to Decision Letter 0]

16 Feb 2021

Response to academic editor and reviewers

Which checklist did you use for your study? 

• COREQ checklist was used and it was attached as an additional information 

Please provide detailed description of the sampling strategy, why you choose the method that you have described and why not any other method of sampling, including rationale for the recruitment method.

• Simple random sampling and purposive sampling were used to select nurses for QAUN and QUAL parts of the study respectively.

• We used simple random sampling because we had sampling frame (list of all the nurses at Jimma medical center. When there is sampling frame, it is better to choose participants using simple random sampling because it produces the representative samples. Systematic random sampling could be applied using this sampling frame but it does not produce samples which are representative as with the method we chose. 

• We selected nurses for QUAL part of the study using purposive sampling method because our intention was to have nurses who could give us detailed information and we had predetermined criteria mentioned in the manuscript. 

Please provide a discussion of potential sources of bias

Researchers work with samples rather than with populations because it is cost-effective to do so. Data from samples can be erroneous and results in sampling bias. Sampling bias refers to the systematic over- or underrepresentation of a population segment. The following is discussion of sources of bias one by one.

• Social desirability bias: This occurs when the study respondents want to place themselves at socially acceptable place. Therefore, the most common source of this bias is respondents’ lack of comfort to reveal their attitudes.

• Recall bias: a systematic error caused by differences in the accuracy or completeness of the recollections retrieved ("recalled") by study participants regarding events or experiences from the past. This occurs when events over the long period of time is asked.

• Self-selection bias or volunteer bias: in studies this offers further threats to the validity of a study as these participants may have intrinsically different characteristics from the target population of the study. Studies have shown that volunteers tend to come from a higher social standing than from a lower socio-economic background.

Please provide a discussion of limitations

• Limitations are provided in the final paragraph of discussion part (Page 18)

Definition of independent and dependent variables are not clear.

• Sorry for its absence in the first manuscript. Dependent and independent variables are operationally defined in the revised manuscript

Does the sample size represent the population of Ethiopia proportionality?

• It does not represent the population of Ethiopia but we used nurses working at Jimma medical center as an accessible population. 

• Ten (10) nurses were interviewed to provide data for QUAL part of the study. As it is obvious, small samples are used for qualitative research. Sample size is based on information need in QUAL research 

Why has p<0.25 used in the bivariate analysis

• In bivariate analysis, independent variables are entered one by one with dependent variable into the bivariate regression model. Then all variables showing significant association in the bivariate analysis are taken together into the multivariable regression model. Variables having insignificant associations at bivariate analysis may show significant association when they are taken together into the multivariable regression model. Therefore, the level of significance should be broadened for bivariate analysis not to miss variables at multivariable analysis. Therefore, p<0.05 leads to missing of variables if used for bivariate analysis. So we used p <0.25 in the bivariate analysis because must literatures recommend this.

---

## [Decision Letter · Decision Letter 1]

3 Jun 2021

PONE-D-20-35734R1

Perceived Transcultural Self-Efficacy and Its Predictors among Nurses in Ethiopia: A Cross-sectional Study

PLOS ONE

Dear Dr. Berhanu,

Thank you for submitting your manuscript to PLOS ONE. After careful consideration, we feel that it has merit but does not fully meet PLOS ONE’s publication criteria as it currently stands. Therefore, we invite you to submit a revised version of the manuscript that addresses the points raised during the review process.

We look forward to receiving your revised manuscript.

Kind regards,

Paola Gremigni, Ph.D.

Academic Editor

PLOS ONE

Journal Requirements:

Reviewers' comments:

Reviewer's Responses to Questions

**Comments to the Author**

1. If the authors have adequately addressed your comments raised in a previous round of review and you feel that this manuscript is now acceptable for publication, you may indicate that here to bypass the “Comments to the Author” section, enter your conflict of interest statement in the “Confidential to Editor” section, and submit your "Accept" recommendation.

Reviewer #1: All comments have been addressed

Reviewer #4: All comments have been addressed

2. Is the manuscript technically sound, and do the data support the conclusions?

Reviewer #1: Yes

Reviewer #4: Yes

3. Has the statistical analysis been performed appropriately and rigorously? 

Reviewer #1: Yes

Reviewer #4: Yes

4. Have the authors made all data underlying the findings in their manuscript fully available?

Reviewer #1: Yes

Reviewer #4: Yes

5. Is the manuscript presented in an intelligible fashion and written in standard English?

Reviewer #1: No

Reviewer #4: Yes

6. Review Comments to the Author

Reviewer #1: Thank you for addressing the comments. It would have been better if a sentence in the abstract was added describing what the score meant. The revised manuscript is okay from my end.

Reviewer #4: There is a great improvement in the revised manuscript as compared with the first one. However, some part of the revised manuscript requires a minor modification.

Title: requires modification. Hence the study was conducted at one Hospital (JMC), is it possible to say “nurses in Ethiopia”?

The word “predictor” is not appropriate for a cross-sectional study. it indicates a great causal association specially used in case of cohort study.

Sample size: there is no any statement in the method part of your manuscript which describe the way of sample size determination for appropriate representation of the general population.

Operational Definitions: what is your reference for your cut of point to say low, middle, or high for TSE?

7. PLOS authors have the option to publish the peer review history of their article (what does this mean?). If published, this will include your full peer review and any attached files.

Reviewer #1: No

Reviewer #4: No

---

## [Author Response · Author response to Decision Letter 1]

15 Jun 2021

Response to Academic editor and Reviewers?

Response to academic editor

 I have added some references (Number 16 & 23), Number 16 has been added because I got important information. Number 23 has been added to cite operational definition which was wrongly left.

 Another changes to the reference list is that it has been made complete and correct as per Vancouver referencing style. Some of the references lacked journals, some lacked page numbers, and some lacked volume and issue number. For some, authors’ last name was wrongly written. I reviewed these things and have corrected them. In addition, I have added DOI for many references. 

Response to reviewers

It would have been better if a sentence in the abstract was added describing what the score meant

 I hope this was raised for the statement which talks about transcultural self-efficacy score in the abstract (The mean transcultural self-efficacy score was 2.89 ±0.59). Even if the result part of the abstract lacked this, it has been described in the conclusion part of the abstract that the score means moderate level of self-efficacy.

Title: requires modification. Hence the study was conducted at one Hospital (JMC), is it possible to say “nurses in Ethiopia”?

 The title was originally “among nurses working at Jimma medical center.” But there was suggestion to say “nurses in Ethiopia” upon previous review. People served at this medical center is very heterogeneous in culture and ethnicity as is the population of Ethiopia. Ethiopia is multi-ethnic country and there is a lot of diversity in culture. People treated at this medical center is also multi-ethnic and diversified in culture. Therefore, we discussed and considered that suggestion because nurses working here at this medical center can represent nurses working in other parts of the country. 

The word “predictor” is not appropriate for a cross-sectional study. it indicates a great causal association specially used in case of cohort study

 I changed this word to the word “associated factors” in the revised manuscript

There is no any statement in the method part of your manuscript which describe the way of sample size determination for appropriate representation of the general population.

 Sorry for the absence. Now I have added sample size determination in the revised manuscript

Operational Definitions: what is your reference for your cut of point to say low, middle, or high for TSE?

 I have added reference in the revised manuscript

---

## [Editor Report · Decision Letter 2]

1 Jul 2021

Perceived transcultural self-efficacy and its associated factors among nurses in Ethiopia: A cross-sectional study

PONE-D-20-35734R2

Dear Dr. Berhanu,

We’re pleased to inform you that your manuscript has been judged scientifically suitable for publication and will be formally accepted for publication once it meets all outstanding technical requirements.

Kind regards,

Paola Gremigni, Ph.D.

Academic Editor

PLOS ONE
---

## [Editor Report · Acceptance letter]

15 Jul 2021

PONE-D-20-35734R2 

Perceived transcultural self-efficacy and its associated factors among nurses in Ethiopia: A cross-sectional study 

Dear Dr. Berhanu:

I'm pleased to inform you that your manuscript has been deemed suitable for publication in PLOS ONE. Congratulations! Your manuscript is now with our production department. 

Kind regards, 

on behalf of

Prof. Paola Gremigni 

Academic Editor

PLOS ONE